# EXPLANATION: Exoplanet and Transient Event Investigation Project—Optical Facilities and Solutions

Gennady Valyavin [1,*], Grigory Beskin [1,2], Azamat Valeev [1,3,4], Gazinur Galazutdinov [1,4], Sergei Fabrika [1], Iosif Romanyuk [1], Vitaly Aitov [1], Oleg Yakovlev [1,5], Anastasia Ivanova [5], Roman Baluev [3], Valery Vlasyuk [1], Inwoo Han [6], Sergei Karpov [1,2,7], Vyacheslav Sasyuk [2], Alexei Perkov [8], Sergei Bondar [8,†], Faig Musaev [1,†], Eduard Emelianov [1], Timur Fatkhullin [1], Sergei Drabek [1], Vladimir Shergin [1], Byeong-Cheol Lee [6], Guram Mitiani [1], Tatiana Burlakova [1,4], Maksim Yushkin [1], Eugene Sendzikas [1], Damir Gadelshin [1], Lisa Chmyreva [1], Anatoly Beskakotov [1], Vladimir Dyachenko [1], Denis Rastegaev [1], Arina Mitrofanova [1], Ilia Yakunin [1,3], Kirill Antonyuk [1,4], Vladimir Plokhotnichenko [1], Alexei Gutaev [1,2], Nadezhda Lyapsina [1], Vladimir Chernenkov [1], Anton Biryukov [2,9], Evgenij Ivanov [1,8], Elena Katkova [8], Alexander Belinski [9], Eugene Sokov [3,10], Alexander Tavrov [5], Oleg Korablev [5], Myeong-Gu Park [11], Vladislav Stolyarov [1,12], Victor Bychkov [1], Stanislav Gorda [13], A. A. Popov [13] and A. M. Sobolev [13]

1   Special Astrophysical Observatory, Russian Academy of Sciences, Nizhnij Arkhyz 369167, Russia
2   Engelhardt Observatory, Kazan Federal University, Kazan 420008, Russia
3   The Faculty of Mathematics and Mechanics, Department of Astronomy, Saint Petersburg State University, 7-9 Universitetskaya Emb., Saint Petersburg 199034, Russia
4   Federal State Budget Scientific Institution Crimean Astrophysical Observatory of RAS, Nauchny, 298409 Crimea
5   Space Research Institute, Russian Academy of Sciences, 84/32 Profsoyuznaya Str., Moscow 117997, Russia
6   Korea Astronomy and Space Science Institute, 776 Daedeokdae-ro, Yuseong-gu, Daejeon 34055, Republic of Korea
7   CEICO, Institute of Physics, Czech Academy of Sciences, 18200 Prague, Czech Republic
8   Research and Production Corporation "Precision Systems and Instruments", Moscow 111024, Russia
9   Sternberg Astronomical Institute, Moscow M.V. Lomonosov State University, Universitetskij Pr. 13, Moscow 119992, Russia
10  Central (Pulkovo) Observatory, Pulkovskoe Shosse 65, Saint Petersburg 196140, Russia
11  Department of Astronomy and Atmospheric Sciences, Kyungpook National University, Daegu 41566, Republic of Korea
12  Cavendish Laboratory, University of Cambridge, Cambridge CB3 0HE, UK
13  Astronomical Observatory, Institute for Natural Sciences and Mathematics, Ural Federal University, 19 Mira Street, Ekaterinburg 620002, Russia
*   Correspondence: gvalyavin@sao.ru
†   Passed away.

**Abstract:** Over the past decades, the achievements in astronomical instrumentation have given rise to a number of novel advanced studies related to the analysis of large arrays of observational data. One of the most famous of these studies is a study of transient events in the near and far space and a search for exoplanets. The main requirements for such kinds of projects are a simultaneous coverage of the largest possible field of view with the highest possible detection limits and temporal resolution. In this study, we present a similar project aimed at creating an extensive, continuously updated survey of transient events and exoplanets. To date, the core of the project incorporates several 0.07–2.5 m optical telescopes and the 6-m BTA telescope of the Special Astrophysical Observatory of RAS (Russia), a number of other Russian observatories and the Bonhyunsan observatory of the Korea Astronomy and Space Science Institute (South Korea). Our attention is mainly focused on the description of two groups of small, wide-angle optical telescopes for primary detection. All the telescopes are originally designed for the goals of the project and may be of interest to the scientific community. A description is also given for a new, high-precision optical spectrograph for the Doppler studies of transient and exoplanet events detected within the project. We present here the philosophy, expectations and first results obtained during the first year of running the project.

**Keywords:** astronomical telescopes; photomerty; spectroscopy; transient events; exoplanets

## 1. Introduction

The achievements of recent decades in the development of wide-angle telescopes and large-format image sensors (CCD, CMOC, etc.), including mosaic ones, have made a true revolution in astronomy and astrophysics, giving the green light to extensive studies of billions of events in outer space available for observations from Earth. This produced new fundamental discoveries and gave fresh knowledge about the Universe. Among these are the discovery of planets orbiting the stars other than the Sun (exoplanets) and massive studies of high-energy gamma-bursts from relativistic objects in the Universe. Let us briefly discuss some of these advanced studies.

Thanks to the efforts of the *Kepler* [1] and *Corot* [2] space missions, as well as the successful operation of various ground-based photometric surveys of exoplanets, such as the famous SuperWASP [3] or a lesser known *Kourovka search for planets* [4], we now have a list of several thousand extrasolar planets and candidates. Based on these studies, the basic physical properties of a large proportion of the confirmed planets are known. Exoplanets cover a wide range of masses, chemical compositions and distances from their host stars. Despite this progress, statistics of the new exoplanet and exoplanet candidates requires more space and ground-based projects aimed at studying the new, as well as the already discovered extrasolar planets and candidates.

The problem of studying the rapidly changing cosmic events from the optical sources (transient events, or transients) was first formulated by H. Bondi in 1970 [5]. To detect and study such sources, wide-angle instruments with high-performance detectors having a temporal resolution of at least a fraction of a second are needed. The latter requirement is due to the short duration (up to 0.01 s) of the considered phenomena. For instance, subsecond highly polarized synchrotron spikes with front durations up to 0.1 s have been first detected in the optical observations of UV Ceti with the 6-m telescope of the SAO RAS [6]. Subsequently, synchrotron flares from red dwarfs were recorded in the millimeter range [7]. To search for and study such events, sufficiently short exposures are required, at least at the level of hundredths of a second, and/or high speeds, up to tens of degrees per second (satellites, space debris, meteors and fireballs).

Nowadays, studies of transients require very wide-angle telescopes in combination with the temporal resolution down to milliseconds. This is due to the advent of a fundamentally new task of searching for optical transients accompanying fast radio bursts [8] and impulses of gravitational waves [9], the duration of which lies in the millisecond range. A number of traditional tasks in the field of optical transient research do also justify the need of bringing the temporal resolution to 0.1 s or below.

In this paper, we present a joint Russian-Korean ground-based astronomical project abbreviated EXPLANATION (EXoPLANet And Transient events InvestigatiON). The project is aimed at a massive photometric, speckle-interferometric, spectral, and radio bolometric search for non-stationary events in the Universe, as well as the study of exoplanets. The core of the project incorporates several 0.07–2.5 m Russian and Korean optical telescopes, as well as the giant 6-m BTA telescope.

In the next section, titled 'What is the EXPLANATION?', we briefly discuss the philosophy of the project and its shareable instrumentation. Then, the 'Design of the main optical facilities' Section presents the details on two main groups of small, wide-angle optical telescopes for the primary detection, a new, high-precision optical spectrograph for the Doppler studies of transient events and exoplanets, and some other classical facilities of the project. The section called as 'Expectations' presents the results we expect from the project. The Results and Discussion sections present the first results obtained in the course of the project and general discussion, respectively.

## 2. What Is the EXPLANATION?

The Explanation project is aimed at both conducting its own ground-based surveys of exoplanets/transient events, and supporting the space missions aimed at the same goals. Its principal difference from the other research is that it integrates a three-level scheme for the diagostics and investigation of non-stationary events, employing a complete range of the instrumentation needed. The scheme consists of a wide-angle optical survey of high temporal resolution (the primary detection level), the second level of examination of the initially detected objects with high angular resolution, and the third level of expert examination of the most interesting objects that have been selected from the first and second levels.

The first level consists of two groups of fast-response wide-angle telescopes with the aperture diameters from 0.07 m to 0.5 m. A detailed description of these instruments is the main goal of this study.

The second level is represented by a group of classic 0.6–1.3 m telescopes. The task for these telescopes is to perform additional studies of the primarily selected targets with higher angular resolution in order to measure their coordinates and multicolor broad-band photometric characteristics with the necessary accuracy.

Finally, the core of the third, expert level of the project involves large Russian and Korean optical telescopes with the apertures ranging from 1.8 to 6 m. The workload of these telescopes is to perform expert photometric, polarization, spectral and speckle-interferometric studies of objects selected during the previous stages. This level also assumes the study of targets from other missions, including the space-based, as already mentioned above.

## 3. Optical Design of Project Facilities

In this section, we describe two groups of wide-angle robotic telescopes of the first level of research—a nine-channel Mini-MegaTORTORA (MMT-9) survey telescope, consisting of nine 0.07-m telescopes (channels), and a group of three+ 0.5-m 2 square degree telescopes.

### 3.1. MMT-9

The Mini-MegaTORTORA (MMT-9) is a wide-field optical monitoring system with high temporal resolution [10] built for and owned by the Kazan Federal University. It has been operated since 2014 under an agreement between the Kazan Federal University and Special Astrophysical Observatory, Russia.

MMT-9 is a successor to the simpler single-channel FAVOR and TORTORA cameras that were operated between 2004 and 2014. Their primary task was an untriggered search for the optical components of gamma-ray bursts, and it proved successful with the discovery of a bright and rapidly variable optical emission from GRB 080319B [11] by the TORTORA camera. The design of MMT-9 builds on this experience, and is aimed towards both increasing the simultaneously observable sky area, maintaining the high temporal resolution, and allowing for a multicolor mode of observations.

All these requirements are solved by the current Mini-MegaTORTORA implementation as a set of nine individual channels installed in pairs on five equatorial mounts. Every channel has a celostate mirror mounted before the Canon EF85/1.2 objective for rapid (faster than 1 s) adjustment of the objective direction within a limited range (approximately 10 degrees in any direction). This allows for either mosaicing a larger field of view, or for pointing all the channels in one direction. In the latter case, a set of color (Johnson's B, V or R) and polarimetric (three different directions) filters may be inserted before the objective to maximize the information acquired for the observed region of the sky (performing both the three-color photometry and polarimetry). High temporal resolution is ensured by using Andor Neo sCMOS detectors with $2560 \times 2160$ pixels 6.4 μm each, providing the $9 \times 11$-degree field of view for every channel, and allowing it to operate with the exposure times as short as 0.1 s.

The implementation of this original operating mode of the MMT-9 system is provided by a two-level scheme of its positioning and maintenance. The first level uses a traditional clock mechanism based on the EQ-6 equatorial mount with a two-coordinate retargeting speed of about 1°/s and a tracking accuracy of about 3″ on a scale of 20 min. On the second level, a quick change of the field of view of each MMT-9 channel is carried out in the range of ±8 degrees clockwise, and ±16 degrees in declination by moving the aforementioned celostate mirror.

It is mounted on a gimbal suspension, and is capable of rotating about two axes with the help of two TGY-MG958 servos. During the observations, the mirror is clamped by brake magnetic shoes and the drive is de-energized. When changing the observed field, power is supplied to the brake coils and the steering machine. The brake coil has two poles and the brake shoe has two mating poles. As a result, at the time of power supply, the shoe is thrown away by the magnetic field, and the mirror is moved to the desired angle by the servo drive. After this operation is completed, power is removed from the machine and the brake, the brake shoe is pulled back, the mirror is braked, and a new observational process begins. A feature of the celeste design is the manufacture of the articulation axes of the gimbal frames on the steel tapes. To attenuate the current surge and save the moment when transferring the mirror in the power circuit drives, the afterburner tanks of a large denomination are provided. For more details, see [12].

Figure 1 shows a general view of two channels with open covers, allowing to see the mirrors of the integral units (the assembly elements of their frames are shown in the top plot of Figure 2). The EQ-6 equatorial mount is in the center between the channels. On the continuation of its hour axis, a switching and ventilation unit is placed, containing power supplies and switching boards. An air duct is located in the mount support, purifying and dehumidifying (the camera) by means of a climate control system, the air that ventilates the housing optical channels (Figure 3). The MMT-9 system, together with the shelter is shown in Figure 4.

The instrument as a whole allows for simultaneous observations of up to 30 × 30 degrees of the sky. Typically, the exposures of 0.1 s (10 frames per second, with an effective detection limit of about V = 11 mag) are used during the monitoring. A dedicated fast differential imaging pipeline allows for a real-time analysis of the data in this mode in order to detect and characterize rapidly variable objects on sub-second time scales and optionally initiate the follow-up observations.

Apart from the high temporal resolution monitoring, the MMT-9 also performs a routine photometric sky survey with a low temporal resolution, only with a deeper limiting magnitude (down to about V = 13.5 mag). More than 1.7 million images, covering every point of the northern sky 5000–20,000 times with the exposures of 20 to 60 s were acquired in this mode.

All these images are catalogued and also analyzed by a dedicated photometric pipeline that processes and calibrates them to the Johnson V band by deriving the photometric equation for every individual image, and then statistically regressing for individual object colors, assuming that their brightness variations are uncorrelated with the changes in the atmospheric conditions. About 30 billion photometric measurements are stored in the database and may be queried by position to extract the light curves of any object detectable by the MMT-9 (The database of images and photometric measurements of the MMT-9 Sky Survey is publicly available at http://survey.favor2.info/).

The accuracy of photometric measurements reachable on the MMT-9 is limited due to very broad point spread function (PSF), and strong pixel-to-pixel inhomogeneity of the CMOS chips of the detectors. On average, the light curve scatter of brighter non-variable objects is about 0.02 magnitudes.

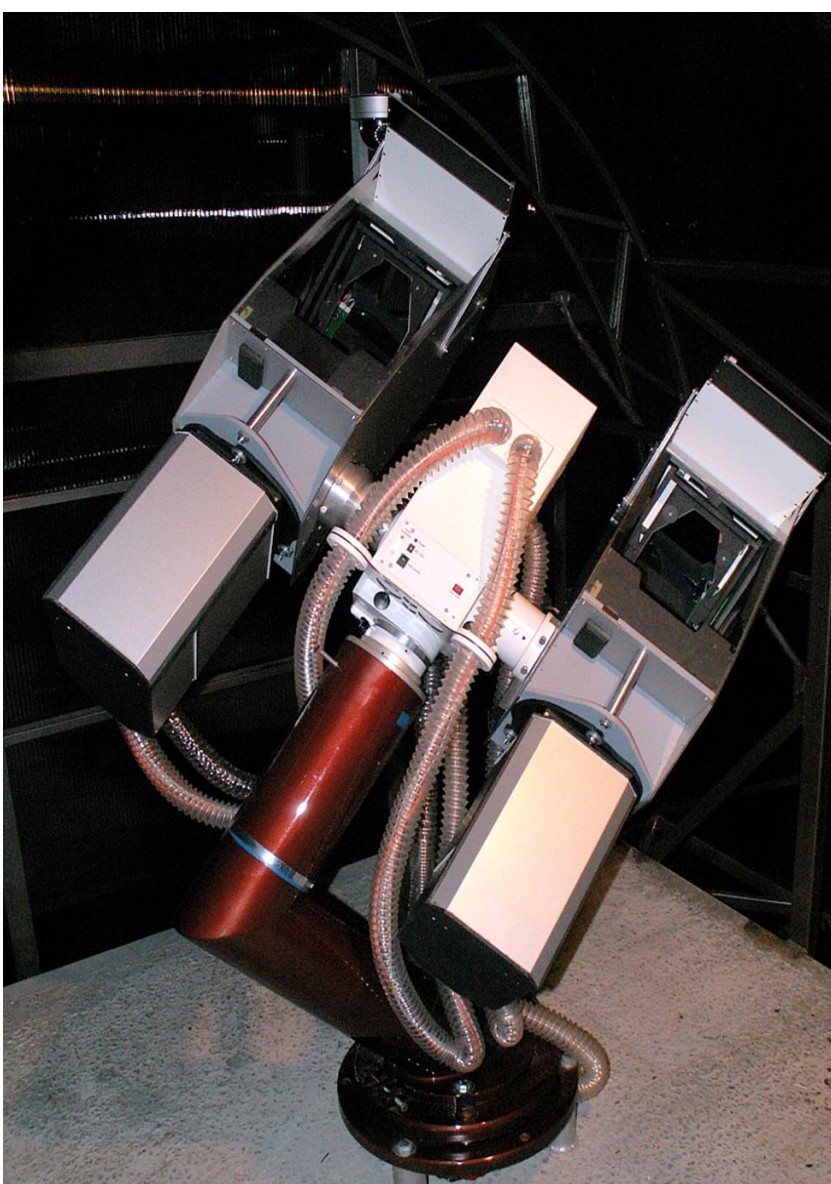

**Figure 1.** General view of the mount with two channels.

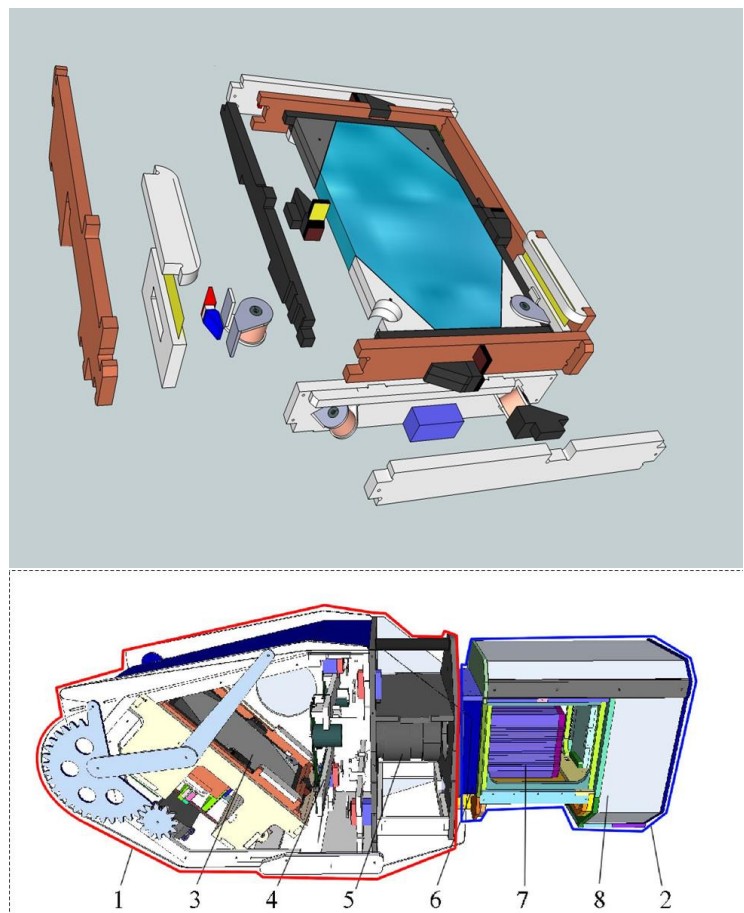

**Figure 2. Top panel**—celostate mirror frame components. **Bottom panel**—a scheme of the optical channel of MMT-9: 1. Unit of celostate mirror and filters. 2. Light receiver. 3. Mirror of the celostate block. 4. Block of filters. 5. Lens. 6. Flange for fastening the chamber block. 7. Light detector. 8. Commutation compartment.

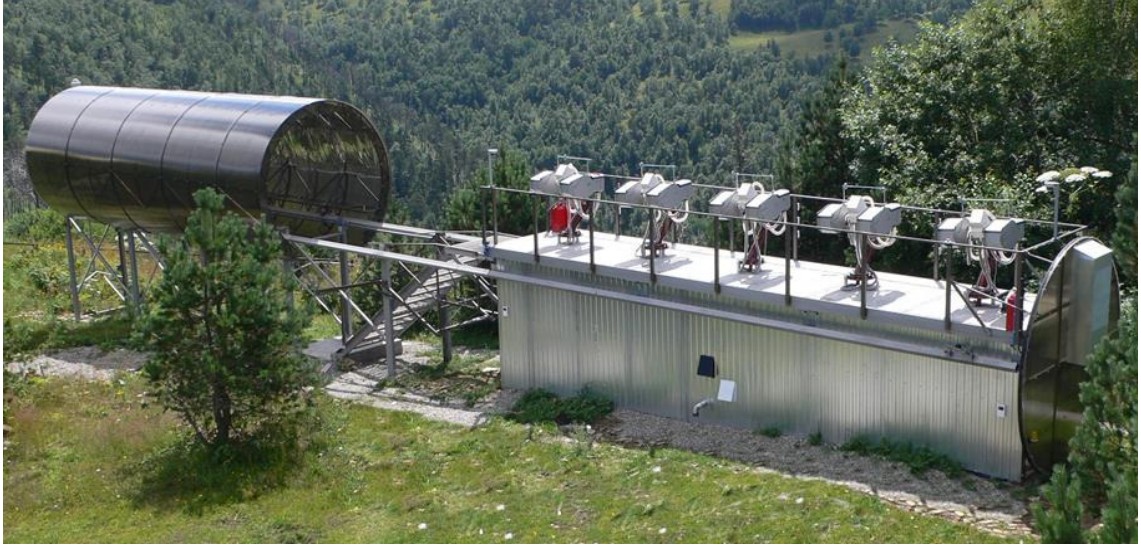

**Figure 3.** General view of the assembled entire Mini-MegaTORTORA system.

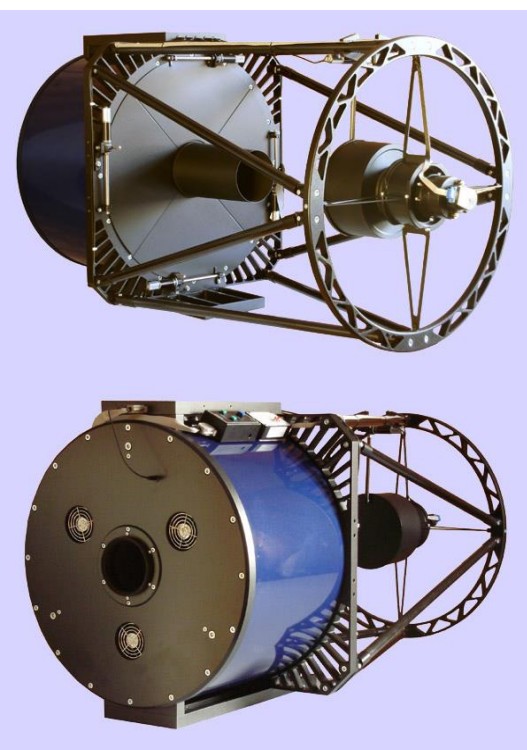

**Figure 4.** General view of the 0.5-m telescope of the array.

### 3.2. Array of 0.5-m Telescopes

In addition to the MMT-9, the survey level of the project is performed by an array of several 0.5 m robotic telescopes. The operation of the array is organized in a way similar to the MMT-9 and was started at the SAO RAS in 2019 by a consecutive development of the first AS500 telescope followed by the second and third ones (fall 2021). Additionally, we plan to install still more (three or more) telescopes with the same apertures over the next few years. Three completely automated telescopes are already operational.

The 0.5-m telescopes we use are designed and manufactured by ASTROSIB (Novosibirsk, Russia). These are classic Ritchey-Chrétien telescopes with the main mirror 0.5 m in diameter. This element is still the same for all the telescopes in the array. Slight differences begin from this point onwards: telescopes No. 1 and 2 are equipped with robotic mounts (10 Micron Mount GM 4000 HPS) made in Italy. Telescope No. 3 has a mount manufactured by Astrosib. The general view of the telescope is presented in Figure 4.

There are also some differences in the design of the 'all-sky' telescope domes. Namely, telescope No. 1 is equipped with a dome manufactured by Baader (Germany), while Nos. 2 and 3 have domes manufactured by Astrosib.

Originally, we have chosen an optical design for the telescopes that provides a maximum field of view of about 1.5 degrees in diameter, which implies using a lens corrector and installing a light detector in the primary focus, where the aperture ratio F/2.7 is implemented. In the future, for a number of telescopes, detectors will be installed in the position commonly used in the Ritchey-Chrétien systems, the Cassegrain F/8 focus. The purpose of this replacement is to improve the scale of the image for precise tasks.

The telescopes are focused by means of standard telescope focusing devices. The large format cameras are manufactured by FingerLake Instrumentation (USA) and based on the CCD devices featuring a 4096 × 4096 KAF16803 front-illuminated array with pixels sized 9 micron. The photometric system is formed by 50 mm turret filters with 5 positions for Johnson's broadband filters. Observational data are recorded in standard 4152 × 4128 16-bit FITS files.

Currently, the array operates independently of the MMT-9 photometric complex, mainly focused on the search for new exoplanet candidates (the first results are outlined

below). However, in the near future, the 0.5-m telescopes along with the MMT-9 will become part of a unified robotic photometric complex controlled by a single software package. The general view of the array is presented in Figure 5.

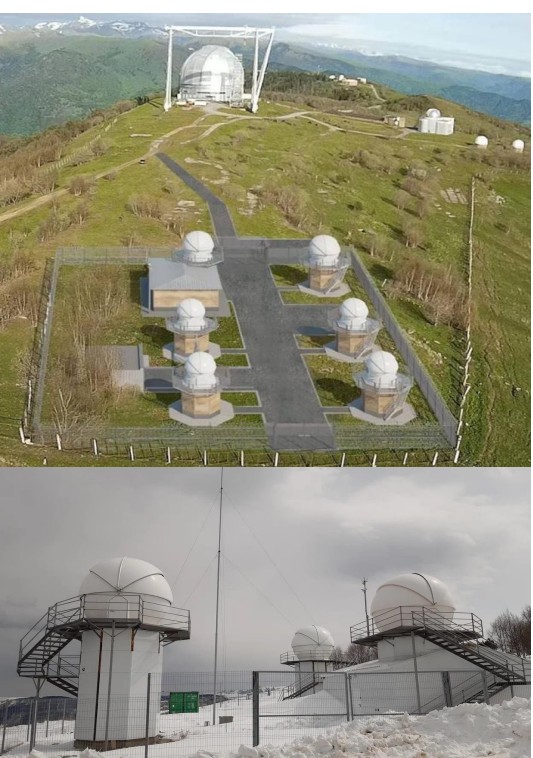

**Figure 5.** General view of the array. **Top plot**: a project. **Bottom plot**: the three operating telescopes.

### 3.3. A Fiber-Fed High Resolution Planetary Spectrograph

In this chapter, we present a scheme of a high-precision fiber-optic spectrograph operating in combination with the 6-m BTA telescope of the SAO RAS. The need to create such an instrument is directly related to the main goals of the project: the study of exoplanets and the diagnosis of bright transient sources selected in the course of the photometric surveys presented above and other studies.

A description of the general concept and optical layout of the spectrograph can be found in Valyavin et al. [13–16]. In our solution, we followed the classical scheme known as the "white pupil" [17]. In contrast to the traditional schemes, the white pupil design uses two off-axis collimators. One of the collimators operates with the echelle grating in the quasi-Littrow configuration, and the other collimator forms the pupil plane at its focus by constructing an undispersed image of the echelle grating there. This is where the cross-dispersion unit and then the focusing optics with the CCD are accommodated. The main advantage of the selected spectrograph layout is its compactness. In this version, the cross-dispersion unit is located exactly in the pupil plane, thereby making it possible to minimize the size of this unit and the focusing camera, and substantially reduce the cost of the entire instrument.

The spectrograph's optical design is presented in Figure 6. The off-axis mirror collimators ((2) and (5) in Figure 6) have parabolic-shaped surfaces with a focal distance of 2175 mm. Echelle grating (3) consists of a mosaic of two standard echelle gratings, each with a blaze angle of 76° (an an R4 grating). The cross disperser (6) consists of a prism made of the OHARA PBMy glass, and a diffraction 300 lines/mm structure at the exit end. The CCD detector (8) is a 4K × 4K camera with a pixel size of 15 microns (http://www.e2v.com). We use an F/2 lens combination consisting of six spherical optical elements with an effective focus of 470 mm as a focusing camera (7).

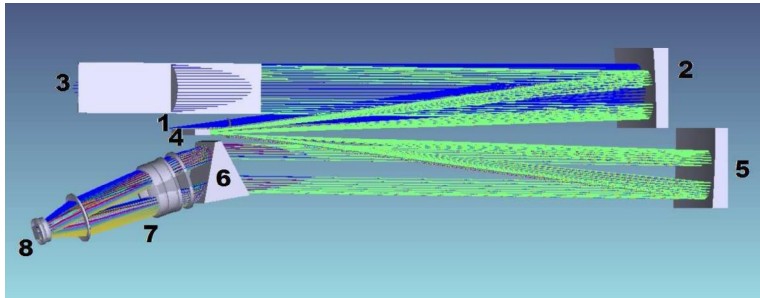

**Figure 6.** Optical scheme of the spectrograph.

At present, the focusing camera is still under construction. However, the availability of simpler standard lenses that can still be used in combination with our spectrograph, resulting in the loss of about 1 stellar magnitude allowed us to start regular scientific observations at the spectrograph in combination with the 6-m telescope. In these conditions, the configuration of the instrument at R = 50,000–65,000 provides observations of a 10–11 magnitude star under the normal weather conditions (the seeing of 1."5 at the SAO RAS) with hour-long exposures and the signal-to-noise ratio S/N = 100.

By using a 63-m fiber assembly, which feeds the light from the star accumulated in the prime focus of the 6-m telescope to a special thermally and mechanically isolated room with the spectrograph (see Valyavin et al. [16] for details), the presently reached accuracy of the radial-velocity measurements for cool stars is up to 3 m/s. In the final configuration, the spectrograph is planned to be equipped with an interferometric control system based on a vacuumized and stabilized FabryPerot interferometer for yet higher, up to 50 cm/s accuracies. The first scientific results with this instrument have already been presented in [18,19].

### 3.4. Other Facilities

The second and third levels in the project's hierarchy are covered by a group of optical telescopes with the apertures ranging from 0.6 and higher. The most significant of them are listed below.

*The Zeiss-1000 telescope* of the SAO RAS with a primary mirror diameter of 1 m. This is a classical optical telescope with a focal ratio of F/13. Its unvignetted field is 45 arcminutes wide, and the typical angular resolution under the North Caucasus weather conditions is about 1.5 arcseconds. The instrument is equipped with a complete range of photometric/polarimetric equipment for the expert diagnostics of transient events and exoplanet candidates detected in the course of the wide-field search described above.

*The AZT-11 Astronomical Reflecting Telescope* of the Crimean Astrophysical Observatory (CrAO). This is a 1.25-m Ritchey-Chretien reflector with a focal length of 16 m, two focuses and all the necessary equipment: a 5-channel photopolarimeter developed by V. Piirola and installed in the main focus and a CCD-photometer at the auxiliary focus. The telescope is basically used for the photometric and polarimetric observations of various space objects: variable stars of different types, active galactic nuclei, exoplanets, asteroids, comets, etc.

*The RC600 0.6 m telescope* of the Caucasian Mountain Observatory (CMO) of Moscow State University (SAI MSU) is a Ritchey Chretien telescope with a 600 mm main mirror diameter and a focal length of 4200 mm, manufactured by ASA (Austria). The instrument is installed on a German parallactic mount ASA DDM160 with direct drive engines with absolute encoders; the ScopeDome 55M slit dome is used as a covering. The RC600 is used for precise photometric observations of exoplanet transits. The photometric unit is equipped with an FLI CenterLine double filter wheel for installing eight 50 × 50 mm filters. The following sets of photometric filters, made using the interference technology are available: U, B, V, Rc, Ic, g′, r′, i′, and Clear glass. The receiver is an Andor iKon-L BV CCD camera, 2048 × 2048 pixels, with a pixel size of 13.5 microns.

Additionally, several 1-meter class telescopes of the Kourovka Observatory (Russia) and the Korea Astronomy and Space Sci. Institute (KASI, Korea) are involved in the photometric part of the project. All these instruments have all the necessary tools for conducting high-precision photometric studies of identified transient events and transits of exoplanets, including large super-Earths—see, for example, the observation of the super-Earth HD219134-b carried out recently with the SAO RAS 1-m telescope [20].

Finally, the top, expert level of the project consists of the following optical telescopes: the 1.8-m telescope of the Bonhuynsan astronomical observatory of KASI (Korea), the CMO 2.5-m telescope, and the SAO RAS 6-m telescope. Instrumentation of these telescopes used in the context of EXPLANATION will be soon described in detail in an another paper.

## 4. Operating Modes

Table 1 provides a brief overview on all the telescopes described above and those already functioning in the project. The first column of the table indicates the name of the telescope, the second gives the size of its aperture, the third presents the field of view, the fourth lists the angular resolution, while the fifth column gives the number of working channels of the telescope (in fact, the number of telescopes in the MMT-9 and the 0.5-m array). We can observe that the telescopes cover completely different fields of view and angular resolutions. However, all of them are aimed at solving one problem—the search and a comprehensive study of transient events and exoplanets.

**Table 1.** Explanation telescopes.

| Telescope | Aperture Size | Field of View | Angular Resolution | Number of Channels |
|---|---|---|---|---|
| | m | arcmin | arcsec | |
| BTA | 6 | 1–6 | 0.06–0.6 | 1 |
| 2.5-m CMO | 2.5 | 10–40 | 0.15 | 1 |
| 1.8-m Bohuynsan | 1.8 | 10–40 | 0.1 | 1 |
| AZT-11 | 1.25 | 12 | 0.6 | 1 |
| 1.2-m Kourovka | 1.2 | 90 | 0.15 | 1 |
| ZEISS-1000 | 1.2 | 8 | 0.8 | 1 |
| RC-600 | 0.6 | 22 | 0.5 | 1 |
| 0.5-m Array | 0.5 | 90 | 0.5–1.4 | 3 |
| MMT-9 | 0.07 | $540 \times 660$ | 10 | 9 |

To date, the entire complex is organized for operation in two main modes: the traditional monitoring of the celestial sphere in order to search for the transiting planets and other variable events near stars and galaxies (Basic mode), and an express analysis of a bright flash from one or another transient event registered in any wide-angle systems (Alert mode). Let us now describe both of these modes.

In the Basic mode, everything goes according to the traditional scheme of searching for transiting planets or other periodic variable events. At the first level of the search for such events with the wide-angle photometric systems, the search is carried out for several months in the selected areas of the sky (with the 0.5-m array) and with a continuous scanning of the Northern Sky (with the MMT-9). The accumulated observational material is constantly analyzed in order to search for regular variability in stars and other galactic or extragalactic objects. As soon as a regular variability owing to a periodic transit of an exoplanet across the disk of a star, or for other reasons, is detected, the coordinates of the star and the ephemeris of the event are broadcast from the first level to the higher levels for the subsequent analysis of the event with the expert telescopes. The operation in the Basic mode is schematically represented in Figure 7 by black arrows. The main feature of the mode is that it does not require any urgent study of detected events. However, things are different in the Alert mode.

If a sudden flare is detected during the MMT-9 photometric survey, or upon receipt of a message from any space mission (usually X-ray), the MMT-9 generates two independent alerts to the array of 0.5-m telescopes and to the second-level telescopes, which embark on the photometric monitoring examination of the event. Unfortunately, the telescopes of the second and third levels (except for the BTA, see below) are not always able to connect to the study immediately after receiving an alert. Therefore, their operation in this mode is considered in a long-term context. In the transient event studies, however, expert observations of the first seconds and hours after the onset of a gamma burst are of particular scientific importance. For this purpose, we have provided a special scheme (see the red arrows in Figure 7) with the participation of 0.5-m telescopes and the BTA.

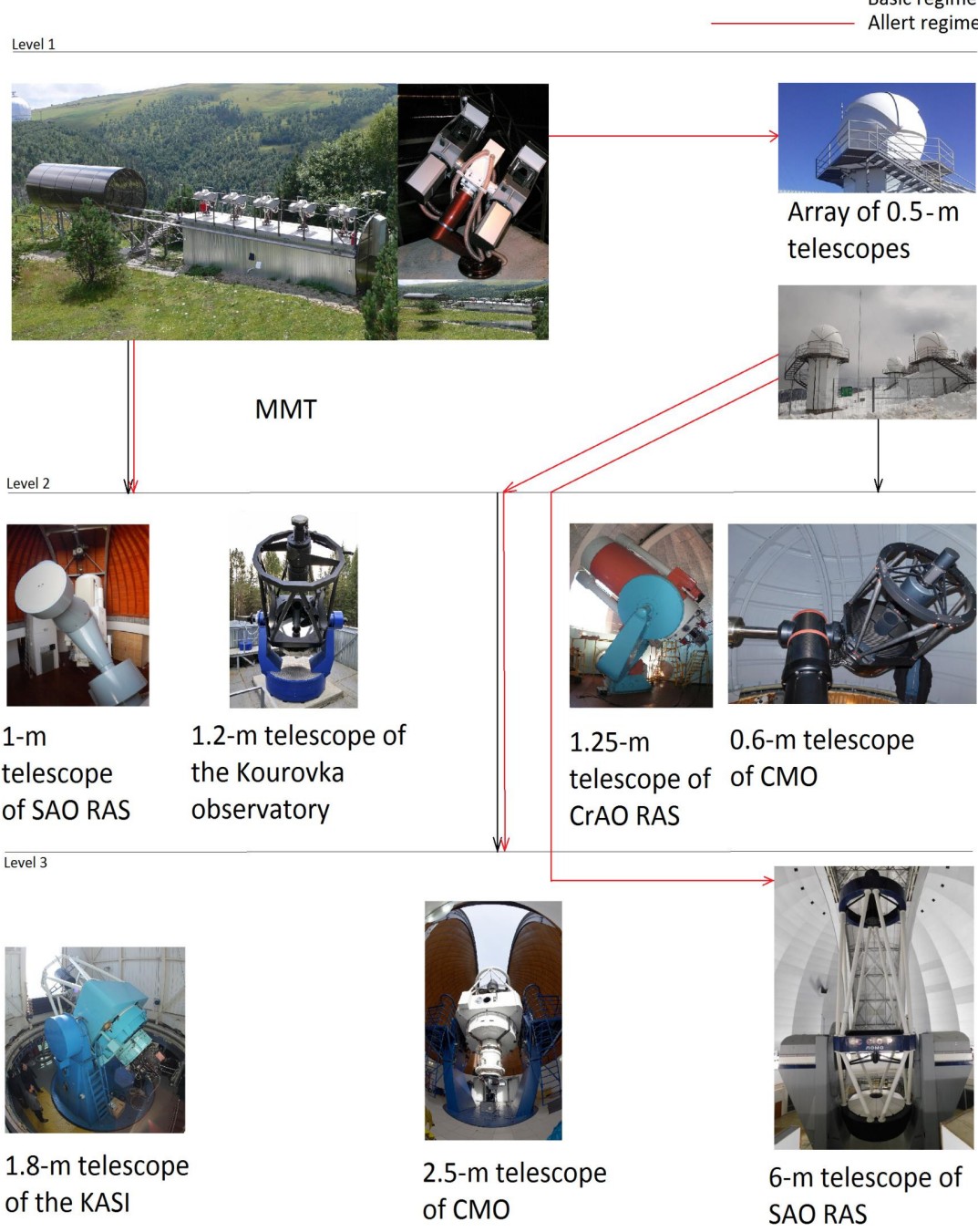

**Figure 7.** Explanation operating modes.

After receiving an alert from the MMT-9, within less than one minute, the 0.5-m telescopes interrupt their operation in the basic mode and embark on targeting at the event, using the roughly generated MMT-9 coordinates in order to refine the coordinates and transmit them to the BTA. The whole process takes 2–5 min, after which the updated coordinates are transmitted to the BTA. There, by agreement with the observer (1 min), the 6-m telescope is pointed at these coordinates and an expert observation of the event starts within 5 to 15 min. The goal of such an elaborate scheme is to start the spectral monitoring of the evolving transient with a large telescope as soon as possible. In our case, we count on the initiation of such observations at times from 8 to 21 min. Employing such a workflow in the astronomical practice is fundamentally new and innovative. We believe this to be a novelty of the Explanation project, in contrast to other projects, where large telescopes within the traditional schemes are usually employed in such studies only with a long delay.

## 5. Expectations

From the scientific perspective, our expectations from the project are mainly based on our experience of several years of MMT-9 service. During the 8 years of MMT-9 operation, it successfully detected and catalogued more than 300,000 meteor events (The database of meteors detected by MMT-9 is publicly available at http://mmt.favor2.info/meteors/) [21], and more than 300,000 passes of 10,000 different artificial satellites (The database of photometric measurements of satellites detected by MMT-9 is publicly available at http://mmt.favor2.info/satellites/) [22], as well as observing the bright prompt optical emission of GRB 160625B [23] and detecting a large number of other transient events, including rapid optical flashes from the glinting satellites [24]. About 9000 flares with a sub-second duration, all belonging to the satellites, have been automatically followed up.

Figure 8 demonstrates the sky coverage (the number of frames covering any given position) of the Mini-MegaTORTORA Sky Survey, as well as the typical light curve scatter for individual stars as a function of their mean magnitude. These figures clearly illustrate the sky coverage efficiency, which we believe to be quite high, 100% of the Northern Sky, and about 10% of the Southern Sky. The bottom plot gives evidence of the effective search for new bright transient events up to $13^m$.

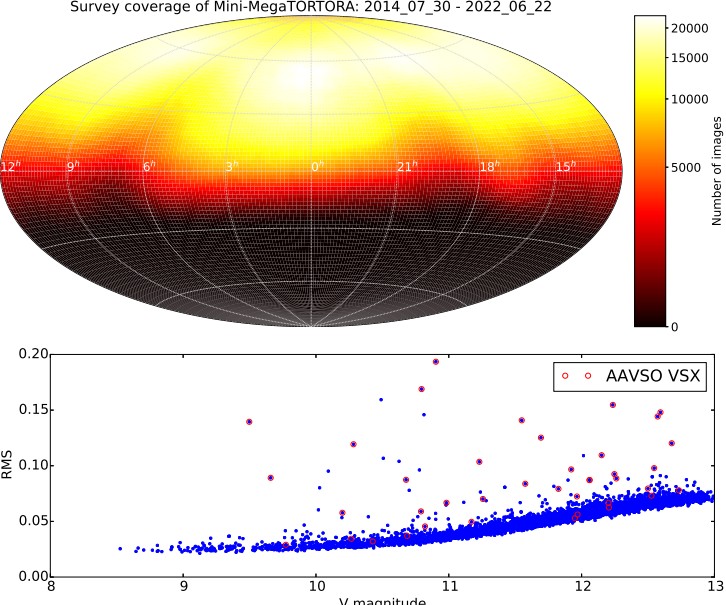

**Figure 8. Top panel**: the sky coverage of the Mini-MegaTORTORA Sky Survey, i.e., the number of survey frames covering a given position in the sky. **Bottom panel**: a typical RMS along the light curves in the survey data as a function of mean magnitude. The red circles mark the known variable stars from the AAVSO VSX database.

The array of the 0.5-m robotic telescopes makes the search for transient events four stellar magnitudes deeper, though, unfortunately, with a 100 times smaller coverage efficiency. Nevertheless, increasing the number of telescopes in the array (only three telescopes are presently in operation) and effective interaction of the array with the MMT-9 within a joint programme planner of the observations shall minimize this drawback.

## 6. Results

In this section, we briefly discuss a couple of examples. Namely, the latest transient and an exoplanet obtained with the observational facilities of the Explanation survey. Please note: Due to the fact that the project has just been started, we cannot yet present in this paper the results exhaustively illustrating the work of the project as a whole in all the operating modes (in particular, the Alert mode). However, the examples below demonstrate the efficiency of the wide-angle telescopic systems and present a new result, discovery of an exoplanet candidate, completely derived within the Basic mode.

### 6.1. Detecting an Optical Flare Accompanying the Extremely Bright Gamma-ray Burst GRB 210619B, and Investigating the Nature of This Event

An optical flare accompanying the gamma-ray burst GRB 210619B [25–28] was detected during the automated monitoring of the celestial sphere in June 2021 by a group of telescopes including the MMT-9. The optical source was registered by the MMT-9 55 s after several space telescopes had simultaneously observed a powerful gamma radiation. The system realigned itself to point towards the region where the gamma-ray source was located after receiving a GCN alert with its coordinates. Observations were carried out synchronously in four channels with the temporal resolutions of 1, 5, 10 and 30 s in the B,V bands and in white light. The scientific details on this discovery are presented in our special study [29].

### 6.2. Searching for Exoplanet Candidates

As part of the search and study of non-stationary events, the project also obtained the first detections of new exoplanet candidate transits. At the time of writing, there are already about a dozen such candidates, detected within the project by the 0.5-m telescope array. A part of these results has been published in [30]. All results will be published in detail after their validation with the 1–2 m and 6-m telescopes. In this paper, we shall limit ourselves to demonstrating one of them, detected by the 0.5-m telescope array, and refined by the 1-m telescope of the second level, Zeiss-1000.

In search for transiting exoplanets, we are looking for and modeling short-term brightness dips in their host stars. Details about this procedure can be found in one of our previous papers (see, for example, [20] and references therein). Briefly, the modeling process is as follows.

The model computes the transit shape in a given spectral band as a function of the relative exoplanet radius expressed as a fraction of the host star radius, equilibrium temperature (Teq) of the exoplanet, the physical characteristics of the star itself, and the impact parameter. The equilibrium temperature of an exoplanet determines the additional intrinsic luminosity when computing the transit depths based on black body relations. Such an approximation is sufficient for the vast majority of the practically realizable cases. The physical properties of the host star (its effective temperature Teff, surface gravity log g, and chemical composition) are used [20] in computations of the linear and squared limb darkening coefficients for a given spectral band.

Figure 9 shows the phase curve of a periodic transit event for a 17th magnitude orange/red dwarf star. The effective temperature of the star is 4957 K and R⋆ = 0.7 solar radii, the corresponding brightness decrease in white light during the transit suggests its exoplanetary nature with a planet of $1.64_{MJup}$. This result has been obtained with the telescopes of different project levels. Once the first positive registration of the transit event was received from the array of the 0.5-m telescopes, other telescopes joined the study as

well. In Figure 9, the results of transit observations obtained with different telescopes are marked by different colors. In particular, black dots illustrate transit observations with the SAO RAS 1-m Zeiss-1000 telescope. The green line is the model of the event.

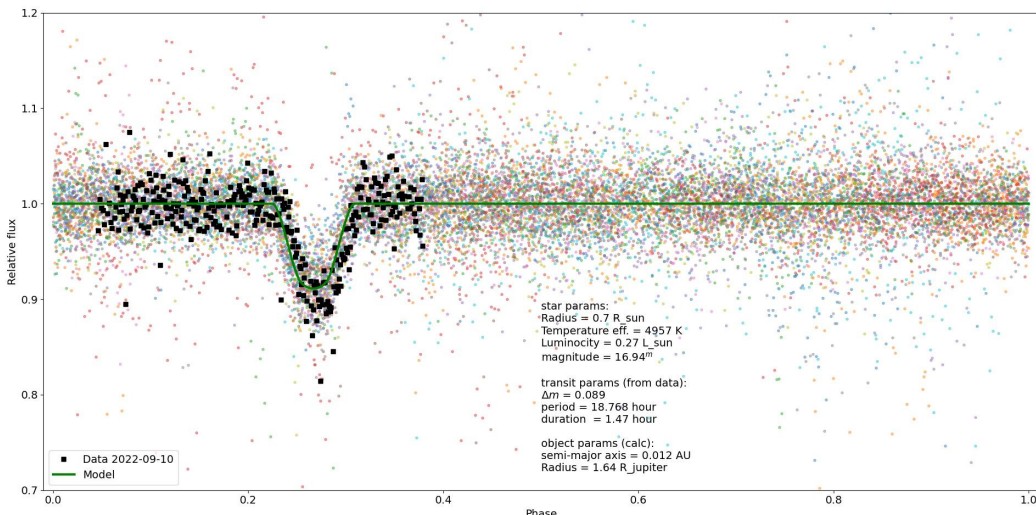

**Figure 9.** An exoplanet candidate detected within the Explanation project.

## 7. Discussion

We have briefly described the new joint Russian-Korean EXPLANATION project, its aims, philosophy, instrumental base, and presented the first results. Generally, the project consists of two large blocks: photometric surveys, and a group of observational facilities and methods for extended studies of objects detected in the course of the surveys. In contrast to the second block, the instrumentation for which has already been elaborated and we have no plans of altering it over the next decade, but the photometric surveys deserve some more discussion.

The configuration of the photometric instruments presented here will be further developed into a more complete network of telescopes with small to medium apertures. In this regard, we also present here (Beskin et al., this issue) one of the ideas for the upgrade. The idea has already been introduced by some of the authors of this paper in [31] within the framework of the SAINT photometric project. We are sure that the upgrade of the wide-field observing facilities as proposed by the SAINT will fundamentally magnify the scientific output of the EXPLANATION project.

**Author Contributions:** Supervision, G.V.; writing—original draft preparation, G.V. and G.B.; conceptualization, G.V., G.B., A.V., G.G., and S.F.; methodology, all authors. All authors have read and agreed to the published version of the manuscript.

**Funding:** The study was funded by the Ministry of Science and Higher Education of Russian Federation, project 075-15-2020-780.

**Institutional Review Board Statement:** Not applicable.

**Informed Consent Statement:** Not applicable.

**Data Availability Statement:** Not applicable.

**Acknowledgments:** The study is supported by the Ministry of Science and Higher Education of Russian Federation, project 075-15-2020-780. MGP was supported by the National Research Foundation of Korea(NRF No-2018R1A6A1A06024970) funded by the Ministry of Education, Korea.

**Conflicts of Interest:** The authors declare no conflict of interest.

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
