# Peer review of "EXPLANATION: Exoplanet and Transient Event Investigation Project—Optical Facilities and Solutions"

_photonics, doi:10.3390/photonics9120950_

Round 1

Reviewer 1 Report

The manuscript is well written. The reviewer think the manuscript can be acceptted providing the following minor suggestions are considered.

1) A table is recommended to clearly show the fucntions and principle parameters of the utilized different kinds of telescopes.

2) Figures of different telescopes are recommended to reordered according to the kinds they belong to.

3) Some notes should be inserted within the figure to make them be easy to be understood.

Author Response

Dear referee, thank you a lot for positive estimaste of our manuscript.

The revised manuscript is attached. All the required modifications in the text due to your comments, and commants of other referees are boldfaced.

 According to your recommendations we have provided the text with a table (Table 1) and a new figure (Fig.7) with all the necessary notices.

Sincerely yours

Gennady Valyavin  

Reviewer 2 Report

I find the paper interesting. It (EXPLANATION project) proposes to join already existing observational facilities, and near future new instrumentation development to study astronomical transient events and the search for exoplanets. I think the approach is the correct one to bring new clues on the specific scientific questions listed above. However, I believe that at the present time, the paper is not ready for publication. I list below a main concern of mine, and subsequent minor comments, that authors should take into consideration.

My main concern is that in the present paper, I didn't find a place where the authors clearly specify how the set of instrumental facilities will work/operate together to fulfill the scientific objectives. I understand that most of them are operating independently of each other, producing and reporting results in the literature. So what EXPLANATION will bring us ? What is the expected additional contribution from EXPLANATION ? how the different observational facilities will be integrated together towards the proposed scientific objectives. This needs to be clearly stated in the paper. I suggest a specific section for this purpose.

For example, results in section 5 are not strictly speaking EXPLANATION project results but rather results obtained by some of its components.

Does EXPLANATION would have provided an earlier (optical) detection of GRB210619B than T0+55 s ?

I understand that RATAN-600 interferometer will be also part of EXPLANATION, but what will be its role in the whole project ?

other minor comments:

line 69: ...the project is not much different from most other similar projects. This a lapalissade. Change or remove

line 117: ...this original mode of functioning of.... use instead ...this original operating mode of....

lines 162-164:  sentence is not clear at all. Please re-write it

line 322: ......nature, with a planet of 1.83Jup.  explain 1.83Jup

Author Response

Dear referee,

Thank you a lot for positive estimate of our paper and very constructive comments. We have taken them all into account. All the corresponding modifications due to your comments, and comments of other referees are boldfaced in the attached revised pdf-version of the manuscript. In particular.

  1. We have removed any mentioning RATAN-600. This is a future of the project. Presently we have focused only at the optical part.
  2. We have added a new section (Operating modes) as you suggested us. 
  3. We have included a new result related to discovery and study of an exoplanet by using a combined photometric study within the Explanation project (not only by an individual instrument).
  4. And we agreed with all your other minor corrections.

sincerely yours,

Gennady Valyavin

Reviewer 3 Report

In this study the authors have described the capabilities of different telescopes involved in a project aiming at the survey of transient events and exoplanets. This is a joint project between Special Astrophysical Observatory of RAS (Russia), other Russian observatories and the Bonhyunsan observatory of Korea Astronomy and Space Science Institute (South Korea). This work summarizes the motivations and the results from this project. Since, the study is mostly describing the instrumentation involved, I could not find much to criticize. The authors have done a good job summarizing different capabilities and giving the background information. I would be happy to recommend this work for publication as it will help the scientific community to become aware about these capabilities. I just have one specific comment:

Figure 8: Can the authors please add description of the model used in Figure 8?

Author Response

Dear Referee,

Thank you a lot for such a high estimate of our paper. We have added a description of the model (green line in Fig.8), provided the text with a reference for detail of the technique we used,  and changed the target. The exoplanet candidate in Fig. 8 is now a new one. The revised manuscript is attached. All modifications in the text due to your comments, and comments of other referees are boldfaced.

sincerely yours,

Gennady Valyavin
